# ECTFE Membrane Fabrication Using Green Binary Diluents TEGDA/TOTM and Its Performance in Membrane Condenser

**DOI:** 10.3390/membranes12080757

**Published:** 2022-07-31

**Authors:** Songhong Yu, Yu Huang, Lixun Zhang, Qian Wang, Zhaohui Wang, Zhaoliang Cui, Enrico Drioli

**Affiliations:** 1State Key Laboratory of Materials-Oriented Chemical Engineering, College of Chemical Engineering, Nanjing Tech University, Nanjing 211816, China; ysh18068848850@163.com (S.Y.); huangyu304@126.com (Y.H.); 201961104005@njtech.edu.cn (L.Z.); chelseawang@njtech.edu.cn (Q.W.); 2National Engineering Research Center for Special Separation Membrane, Nanjing Tech University, Nanjing 211816, China; 3Jiangsu National Synergetic Innovation Center for Advanced Materials (SICAM), Nanjing Tech University, Nanjing 211816, China; 4Research Institute on Membrane Technology, ITM-CNR, Via Pietro Bucci 17/C, 87036 Rende, Italy; e.drioli@itm.cnr.it

**Keywords:** ECTFE membrane, membrane condenser, binary diluents

## Abstract

Poly(ethylene-chlorotrifluoroethylene) (ECTFE) membrane is a hydrophobic membrane material that can be used to recover water from high-humidity gases in the membrane condenser (MC) process. In this study, ECTFE membranes were prepared by the thermally induced phase separation (TIPS) method using the green binary diluents triglyceride diacetate (TEGDA) and trioctyl trimellitate (TOTM). Thermodynamic phase diagrams of the ECTFE/TEGDA: TOTM system were made. The effects of the diluent composition and cooling rate on the structure and properties of the ECTFE membranes were investigated by characterizing the SEM, contact angle, mechanical properties, pore size and porosity. The results showed that ECTFE membranes with cellular structure were successfully prepared and exhibit good mechanical properties. Moreover, increasing the TOTM content in the binary diluents and decreasing the cooling rate could effectively improve the mean pore size of the ECTFE membranes, but the increase in TOTM content reduced the mechanical properties. During the MC process, the water recovery performance of ECTFE membranes increased with the increase in the mean pore size of the membranes, and the condensation flow and water recovery of membrane prepared at 20% TOTM were 1.71 kg·m^−2^·h^−1^ and 54.84%, respectively, which were better than the performance of commercial hydrophobic PVDF membranes in the MC. These results indicated that there is good potential for the application of ECTFE membranes during the MC process.

## 1. Introduction

Chemical plants, cooling towers and power plants always emit large amounts of high-humidity flue gas into the air. Recycling the water is meaningful work which, on the one hand, can alleviate environmental pollution and, on the other hand, can effectively improve the utilization rate of water resources and alleviate the problem of water shortage [1].

Traditional water recovery technologies mainly include: liquid–solid adsorption, low-temperature separation [2] and condensation cooling [3]. Liquid–solid adsorption [4] has the problems of large loss of desiccant and high cost. Low-temperature separation is expensive because of the large difference in boiling point between gas and water. Condensing cooling is the simplest process, but corrosion due to the presence of acid contaminants in the exhaust flow limits its widespread use [5].

With the development of membrane materials and the continuous optimization of performance, membrane separation methods have been applied to recover water vapor from high-humidity flue gas. According to the different separation principles, they can be divided into the following three categories: hydrophilic dense membrane condenser (DMC), also known as vapor permeation membrane; transport membrane condenser (TMC) and membrane condenser (MC) [6,7]. Figure 1 shows the dehydration mechanisms of different membranes. A DMC adopts a hydrophilic dense membrane with water molecule selectivity, and realizes the separation of water vapor in flue gas based on the dissolution–diffusion mechanism. However, since the water vapor is completely limited by pressure in the process, the energy consumption of the process is relatively high. The membrane material for the TMC is mainly ceramic. The principle is that water vapor condenses in the membrane pores, and non-condensable gases cannot penetrate through the membrane pores. Therefore, during operation, not only high-quality condensed water can be obtained, but also the waste heat in the flue gas can be effectively recovered. Although the ceramic membrane has good thermal conductivity, its high price limits its application, and a large amount of cooling circulating water needs to be used to keep the surface temperature of the ceramic membrane constant during operation, which increases the energy consumption in the process and is not conducive to promotion. Macedonio et al. [7] proposed a new water recovery method, a membrane process that can selectively recover water from industrial gases, known as a membrane condenser (MC). The principle is that the feed gas is in contact with the porous hydrophobic membrane, the water vapor condenses on the membrane surface, the hydrophobicity of the membrane is used to prevent the liquid from infiltrating into the pores and the dehydrated gas directly penetrates through the membrane pores.

The MC process requires a high degree of hydrophobicity for membrane materials, as contaminants in the solution can slowly weaken the hydrophobicity of the membrane [8], thus affecting the performance of the MC in the long term. At present, the hydrophobic membrane materials commonly used in the MC process mainly include polyvinylidene fluoride (PVDF) [9], polypropylene (PP) [10] and poly(ethylene-chlorotrifluoroethylene) (ECTFE) [11]. ECTFE is attracting attention due to its corrosion resistance and hydrophobicity. Drioli et al. [12] applied ECTFE membrane to the MC process for the first time, and compared it with commercial PVDF membrane, finding that the two membranes had similar water recovery rate, indicating the application prospect of ECTFE membrane in the MC process. ECTFE is a 1:1 copolymer of ethylene and chlorotrifluoroethylene, which is a polymer with alternating chains, and its fluorine content is about 40%, resulting in good hydrophobicity and excellent chemical resistance [11,13]. As ECTFE is insoluble in any solvent at room temperature, ECTFE membranes are usually prepared by thermally induced phase separation (TIPS).

In the membrane preparation process of the TIPS method, the diluent is one of the key factors in the crystallization process of the polymer [14], which affects the membrane properties such as pore size and mechanical strength. Cui et al. [15] successfully prepared PVDF membranes with good tensile strength using dibutyl maleate (DBM) as a diluent. Compared with some works [14,16], they found that the PVDF/DBM system exhibited a wide liquid–liquid phase separation region, as shown in Figure 2, which facilitated the formation of a bi-continuous structure and led to improved membrane tensile strength. The effects of diluents on the properties and morphology of ECTFE membranes have attracted the attention of researchers. Hamed et al. [17] prepared ECTFE hollow fiber membranes for the first time using diethyl phthalate (DEP) and glycerol triacetate (GTA). They found that in the GTA system, the spherulites were more numerous and smaller in size than in the DEP system. Pan et al. [18] successfully prepared ECTFE membranes with conveniently controlled microstructures using the binary diluents bis(2-ethylhexyl) adipate (DEHA) and DEP, and the cross section of the membrane changed gradually from honeycomb to spherulitic with an increase in the DEHA ratio. Liu et al. [19] found that when trioctyl trimellitate (TOTM) was used as a diluent, a bi-continuous structure resulting from an obvious L-L phase separation could be observed when the ECTFE concentration was 15%. The cooling rate also has an effect on the structure and properties of ECTFE membranes in the TIPS method. Roh et al. [20] found that the mean pore size and 2-propanol permeate flux of the ECTFE membrane decreased as the cooling rate increased. This is because the increased cooling rate shortened the L-L phase separation time and resulted in smaller droplet growth.

In this study, ECTFE membranes were successfully prepared by the TIPS method using the green binary diluents triglyceride diacetate (TEGDA) and TOTM. The effects of diluent composition and cooling rate on the structure and properties of the ECTFE membranes were studied, and the application potential of ECTFE membrane in the MC process was investigated.

## 2. Materials and Methods

### 2.1. Materials

ECTFE (901) was supplied by Zhejiang Chemical Industry Research Institute Co., Ltd (Hangzhou, China). TOTM, TEGDA and kerosene were supplied by Aladdin Biochemical Technology Co., Ltd. (Shanghai, China). GQ-16 was obtained from Gaoqian Functional Materials Co., Ltd. (Nanjing, China). Anhydrous ethanol was supplied by Sinopharm Group Chemical Reagent Co., Ltd (Shanghai, China). Poly(vinylidene fluoride) (PVDF) membrane was purchased from Millipore Co., Ltd. (Bedford, MA, USA). Deionized water was used in all experiments. The molecular structures of ECTFE, TOTM and TEGDA are shown in Figure 3.

### 2.2. Phase Diagram Parameter Measurement

The phase diagram is a powerful means of determining the compatibility between the polymer and the solvent, and can also predict the structure of the membrane. The ECTFE powder and binary diluents were stirred at 220 °C for 3 h to form a homogeneous casting solution, which was allowed to stand for 30 min to degas and then poured into a Petri dish to cool to room temperature. An appropriate amount of casting solution was taken with forceps, placed between a pair of wiped slides and fixed on a hot stage. The hot stage was heated to 220 °C at 30 °C/min and kept at a constant temperature for 5 min until the casting solution became a homogeneous liquid, then cooled down at 6 °C/min. The cloud point was determined by observing the appearance of turbidity in the casting solution by a polarizing microscope (XPV-800E, Shanghai, China). The temperature of the hot table when the casting solution became cloudy was the cloud point temperature of the sample. The crystallization temperature was determined using differential scanning calorimetry (DSC, Q-20, New castle, DE, USA). The sample was placed in an aluminum differential scanning calorimetry pan. The heating rate was 10 °C/min, the initial temperature was 30 °C and the temperature was raised to 220 °C for 5 min and then cooled to 30 °C at a rate of 10 °C/min. The onset of the exothermic peak is the crystallization temperature of the sample.

### 2.3. ECTFE Membrane Preparation

The ECTFE membrane was prepared by the TIPS method. The dried ECTFE powder was mixed with the binary diluent TEGDA/TOTM and stirred at 220 °C for 3 h using a mechanical stirrer to form a homogeneous solution. After degassing at the same temperature, an appropriate amount of the casting solution was poured into a preheated stainless steel mold (around 220 °C). After that, the mold was then placed on a hot press and was pressurized at 220 °C for 5 min. Finally, the mold was quenched in a water bath at the desired temperature. The nascent ECTFE membrane was placed in an ethanol bath to extract diluent overnight, and then the final ECTFE membrane was obtained after freeze-dying. The content of ECTFE was 20% and TEGDA and TOTM together were 80%. Afterwards, when examining the effect of cooling rate, the TOTM content was 15%.

### 2.4. Membrane Characterization

The surface and cross section morphologies of the prepared ECTFE membranes were examined by a cold field emission scanning electron microscope (FESEM, S-4800, Tokyo, Japan). All samples were freeze-fractured in liquid nitrogen and sputtered with gold.

The pore size was measured by a membrane pore size distribution apparatus (PSDA-20, Gaoqian function Co., Nanjing, China). The samples were first cut to the appropriate size and then immersed in the wetting agent GQ-16. After 2 h, the samples were removed, placed in the assembly and the mean pore size was determined by the gas–liquid exclusion method. The porosity was calculated using Equation (1):(1)$$\varepsilon =\frac{\frac{m_{1}-m_{0}}{\rho_{k}}}{\frac{m_{1}-m_{0}}{p_{k}}+\frac{m_{0}}{\rho_{p}}}\times 100\%$$
where ε represents the porosity of the membrane (%); m_0_ represents the mass of the sample before immersion in kerosene (g). m_1_ represents the mass of sample soaked in kerosene for 24 h (g). $\rho_{p}$ representss the density of the sample (about 1.68 g/cm^3^); $\rho_{k}$ represents the density of the wetting medium kerosene (about 0.82 g/cm^3^).

The contact angle (CA) was obtained by measuring more than five different positions of the same sample at 25 °C using the sessile drop method on a contact angle meter (DropMeter A-100, MAIST Measurement Co., Ltd., Ningbo, China).

A tensile testing instrument (HLD 1000, Wenzhou, China) was used to measure the mechanical properties of ECTFE membranes. Each sample was cut into the same shape using a Japanese knife mold, and the thickness of each sample was measured with an electronic digital membrane thickness gauge before testing. Both ends of the samples were fixed and stretched at a constant rate of 5 mm/min (25 °C).

### 2.5. MC Performance of ECTFE Membrane

Figure 4 shows the MC setup used in this study. The performance evaluation system of the MC consists of two parts: condensation flow and water recovery rate. The device was run for 10 h in total, and the change in water recovery performance of the permeable side was recorded every 1 h. The MC operating parameters are shown in Table 1.

The condensation flow can be calculated by using Equation (2):
(1)Condensation flow
(2)$$J=\frac{\Delta m}{A\cdot \Delta t}$$
where *J* represents condensation flow (kg·m^−2^·h^−1^); Δ*m* represents the mass of condensate collected from the retained side (kg); *A* represents the effective area of the used flat membrane (m^2^); Δ*T* represents the membrane condensation operation time (h).


(2)Water recovery rate


The water recovery rate can be expressed by using Equation (3):(3)$$R=\frac{\Delta m}{M}$$
where *R* represents the water recovery rate (%); Δ*m* represents the mass of condensate collected from the retained side (kg); *M* represents the mass of water vapor contained in the feed gas, which can be calculated from a dew point meter (kg).

## 3. Results and Discussion

### 3.1. The Effect of Diluent Composition on ECTFE Membranes

#### 3.1.1. Phase Diagram

In order to know the compatibility between polymer and solvent, Hansen solubility parameter (HSP) theory is used. The HSP distance (*R*) was obtained by calculating the solubility parameters *δ**_d_*, *δ**_p_* and *δ**_h_*, where *δ**_d_*, *δ**_p_* and *δ**_h_* corresponded to the dispersion force, polar force and hydrogen bonding of the substance, respectively, and *R* was evaluated by [21].
(4)$$R^{2}=4{\left(\delta_{d,p}-\delta_{d,d}\right)}^{2}+{\left(\delta_{p,p}-\delta_{p,d}\right)}^{2}+{\left(\delta_{h,p}-\delta_{h,d}\right)}^{2}$$

The *R* values of ECTFE and diluents are shown in Table 2 A smaller *R* value indicates better compatibility of ECTFE and diluent. From Table 2, the *R* value of the ECTFE/TEGDA system is 11.36, higher than in other published work, so we introduced TOTM to improve the compatibility of ECTFE and diluents. In the previous work, good compatibility was demonstrated between TOTM and ECTFE [19], so it can be used to improve the compatibility of the ECTFE/(TEGDA:TOTM) system.

Figure 5 shows the effect of diluent composition on the thermodynamic phase diagram of the system. With the increase in TOTM content, the cloud point temperature of the system showed a decreasing trend, while the crystallization temperature did not change significantly. In the TIPS process, the difference between the cloud point temperature and the crystallization temperature is defined as the liquid–liquid (L-L) [23] phase separation region. In this region, the polymer/diluent system is thermodynamically unstable and small changes in temperature can cause phase separation, resulting in a polymer rich phase and poor phase, but both phases are still liquid at this time. Therefore, as the TOTM content increases, a narrowing of the L-L phase separation region can be observed. When the TOTM content is higher than 25%, the L-L phase separation region disappears, and only solid–liquid (S-L) phase separation occurs during the cooling process, with the polymer crystallizing and solidifying directly from the solution.

#### 3.1.2. Morphology

Figure 6 shows the effect of diluent composition on the surface and cross section structure of the ECTFE membranes. As shown in Figure 6k–o, when the TOTM content was low, the cross section structure of the ECTFE membrane showed a closed cellular structure with a large cellular volume. As the TOTM content increased, the cross section structure of the ECTFE membrane changed from closed structure to bi-continuous structure and the cellular volume became smaller. In the TIPS process, systems with a wide L-L phase separation region tend to form cellular structures or even a bi-continuous structure [15,24]. As the system cools down into the L-L phase separation region in the homogeneous state, there is a tendency for the diluent to migrate into the polymer lean phase, while the polymer migrates to the rich phase (both phases are still in the liquid state). As the system cooled further into the S-L phase separation state, the polymer began to crystallize and solidify, while the remaining diluent that had not migrated to the poor phase migrated and precipitated outside the crystal structure and even to the membrane surface, where the diluent was extracted to form a cellular structure. In the TIPS process, the L-L phase separation of the casting solution system will form a typical cellular structure during cooling [25], but the cellular volume and the connectivity between adjacent cavities are related to the size of the L-L phase separation region. TOTM is a good solvent for ECTFE and the increase in its content enhances the compatibility of the polymer with the solvent, making the system less susceptible to phase separation and leading to a reduction in the L-L phase separation region. The system already enters the S-L phase separation with a short coarsening time of the polymer lean phase growth, and the diluent migrates mainly outside the cellular structure, so the cavities are smaller in size but have increased interconnectivity and large cleavages can be observed at a TOTM content of 20% in Figure 6e.

#### 3.1.3. Mean Pore Size, Porosity and Water Contact Angle

Figure 7 shows the effect of diluent composition on the mean pore size, porosity and water contact angle of ECTFE membranes. With the increase in TOTM content, the mean pore size and porosity of the ECTFE membrane showed an increasing trend. At 10% TOTM content, the mean pore size and porosity of the ECTFE membrane were 46.6 nm and 60.24%, respectively. After a 10% increase in TOTM content, the mean pore size and porosity increased to 198 nm and 74.71%, respectively. The pore size of the ECTFE membrane increased by more than three times. In the TIPS process, the formation of membrane pores is mainly dependent on diluent migration during the cooling of the homogeneous system, which occurs in two ways. One is that when the interaction between the polymer and the diluent in the system is strong, the system cools down and passes through the L-L phase separation region before undergoing S-L phase separation. The system remains in solution in the L-L phase separation region, but is split into a polymer-lean phase and a polymer-rich phase. The diluent mainly concentrates into the lean phase to form larger diameter pores, while the polymer-rich phase grows to form a continuous phase membrane matrix. When the system enters the S-L phase separation region, the polymer begins to crystallize and precipitate from the diluent, which means that the diluent is repelled, squeezed out of the crystalline structure, and migrates outward, thereby penetrating the adjacent cellular structure. Another way is that with the increase in TOTM content in the system, the interaction between the polymer and the diluent is weakened, the compatibility of the system is enhanced, the L-L phase separation region is narrowed and the diluent migrates outward before it can enrich into the polymer-lean phase, forming the “channels” between adjacent luminal structures. Therefore, it can be observed that the mean pore size and porosity of the membrane exhibit a positive correlation trend with the TOTM content in the binary diluents.

The surface properties of the membrane have a non-negligible effect on the condensation of moisture from the flue gas on the membrane surface. As shown in Figure 7, the contact angle did not change much and remained around 130°. The is because the introduction of the good diluent TOTM mainly changes the interaction force between the polymer and the diluent, which affects the phase separation process of the casting solution during the cooling process. The surface morphology of the ECTFE membranes produced with different TOTM contents also showed that the TOTM content had little influence on the surface morphology of the membranes, so the surface hydrophobicity of the ECTFE membranes was less influenced by the TOTM content.

#### 3.1.4. Mechanical Properties

Figure 8 shows the effect of diluent composition on the mechanical properties of ECTFE membranes. With the increase in TOTM content, the tensile strength and elongation at break of the ECTFE membrane showed a downward trend. As shown in the cross section SEM images in Figure 6, with the increase in TOTM content, the cross section structure of the membrane changed from a closed cellular structure to a penetrating bi-continuous structure, and cracks appeared when the TOTM content was 20%. Due to the thinning of the wall thickness, the adjacent cavities are not tightly connected, resulting in a reduction in the mechanical properties of the membrane. The ECTFE membranes prepared by further increasing the TOTM content were brittle and difficult to remove from the mold, so this experiment only investigated the increase in the TOTM content to 20%. Moreover, compared to some previous work, the mechanical properties of the prepared ECTFE membranes in this study are better. Due to the introduction of TEGDA, which further improves the compatibility of the polymer with the solvent and produces a homogeneous cellular structure, the cellular structure of the membrane usually exhibits better mechanical properties [26,27].

#### 3.1.5. MC Performance

Figure 9 shows the performance of ECTFE membranes made with different diluent compositions during the MC process. The water recovery rate and condensation flow of ECTFE membranes during the MC process are very low when TOTM content is low. This is because the pore size of ECTFE membranes is very small at low TOTM content, far below the requirements of the MC process for a hydrophobic membrane. The small pore size of the membrane causes the feed gas to stagnate on the surface of the membrane, making it difficult for it to pass through, and the pressure inside the module is too high, causing damage to the hydrophobic membrane and the module. As the TOTM content increased, the pore size of the ECTFE membranes increased and gradually met the requirements for the MC. The ECTFE membranes prepared at 20% TOTM showed high recovery performance during the MC process, achieving condensation flux of 1.71 kg·m^−2^·h^−1^ and water recovery rate of 54.84%, better than commercial hydrophobic PVDF membranes, demonstrating the potential of ECTFE membranes in the MC process. Meanwhile, compared to the previous work of the group [28], this study further changes the membrane preparation method to improve the water recovery rate.

### 3.2. The Effect of Cooling Rate on ECTFE Membranes

#### 3.2.1. Morphology

Figure 10 shows the SEM of the ECTFE membranes prepared at different cooling rates. As shown in Figure 10a–e, with the decrease in the cooling rate, the morphology of the membrane surface changed from a fissure-like pore structure to a rough, gully-like structure. During the cooling of the system, although the faster cooling rate is beneficial to the rapid nucleation of polymer crystals, it also shortens the crystal growth time. When the cooling rate is slower, the system enters the L-L phase separation region for a longer time, the growth and coarsening process of the poor polymer phase is more complete, the diluent tends to be enriched in the polymer-lean phase, the pore volume becomes larger and the pore wall becomes thicker [29,30]. When the system enters the S-L phase separation stage, the residual diluent permeates out of the continuous phase membrane matrix, forming a sponge-like structure. In addition, the main reason for the change in the surface morphology of the ECTFE membranes is the decrease in the cooling rate, the longer the system enters the L-L phase separation region, the tendency for the diluent to enrich in the polymer-lean phase, the high surface polymer content and the agglomerative build-up forming a raised structure.

Figure 11 shows the changes in the mean pore size and porosity of ECTFE membranes produced at different cooling rates. As the cooling rate decreased, the mean pore size of the ECTFE membranes showed an overall increasing trend, while the porosity remained basically the same at around 65%. In the TIPS method, both the widening of the L-L phase separation region and the extension of the system into the L-L phase separation region increased the volume of the cell lumen, and the decisive factor in determining the interconnectivity of the cell lumen is the outward migration of the solvent. At lower cooling rates, the polymer crystallizes more slowly and the solvent exclusion process is slower, forming sponge-like pore walls from the continuous phase polymer matrix and increasing the membrane pore size. Figure 11 also shows the variation in water contact angle on the surface of the ECTFE membrane produced at different cooling rates. The contact angle changed a little, remaining at about 135 °. This indicates that the change in cooling rate has less effect on the hydrophobic properties of the membrane surface.

#### 3.2.2. Mechanical Properties

Figure 12 shows the changes in the mechanical properties of the ECTFE membranes prepared at different cooling rates. As the cooling rate decreased, the tensile strength and elongation at break of the ECTFE membranes showed an overall slow decrease. At 0 °C, the tensile strength and elongation at break of the ECTFE membranes reached a maximum of 2.88 MPa and 14.51%, respectively, while at 60 °C, the tensile strength and elongation at break decreased by 0.59 MPa and 7.83%, respectively. In the TIPS method, the reduced cooling rate increases the growth coarsening time of the polymer-lean phase and increases the volume of the cell lumen. Although thickening the pore walls, this will encourage diluent to seep out of the pore walls during polymer curing to repel the diluent, producing a spongy structure with weaker mechanical properties.

#### 3.2.3. MC Performance

Figure 13 shows the performance of the ECTFE membranes in the MC at different cooling rates with the same operating parameters as Table 1. The cooling rate is slowest at 60 °C, when the water recovery performance of the membrane is at its highest, with a condensation flow and water recovery rate of 0.52 kg·m^−2^·h^−1^ and 16.74%, respectively. This is mainly due to the fact that the mean pore size of the membrane gradually increases as the cooling rate decreases. It is worth noting that when the cooling rate reached its minimum value, the water recovery performance increased significantly, but the pore size growth did not show a corresponding trend. The surface morphology and hydrophobic properties of the membrane affected the condensation process of water from the flue gas on the membrane surface during the MC process. As mentioned earlier, when quenching baths with different temperatures were used, the surface structures of the membranes were found to be different, which also indicated that tuning the surface structure of the membranes was an effective means to improve MC performance during the MC process.

The data for some membrane condensers are listed in Table 3. As the membrane types were not exactly identical and there were differences in the operating parameters, our work showed different results of MC performance. In Pan et al.’s work [28], the condensation flow is 1.1–1.8 kg·m^−2^·h^−1^, while ours is 0.42 kg·m^−2^·h^−1^, but the feed flow rate in their work is two times higher than ours. A higher condensation flow can be achieved at a higher feed flow rate [31]. The advantage of our work is the use of less toxic diluents to prepare ECTFE membranes that can be used in the MC process.

## 4. Conclusions

In this paper, ECTFE membranes were prepared using the TIPS method with a binary environmentally friendly diluent. The effects of diluent composition and cooling rate on the membrane structure and properties were investigated. The prepared ECTFE membranes with cellular structure showed good mechanical properties. As the TOTM content increased, the cross section structure changed from a closed cellular structure to a bi-continuous one. The change in cooling rate mainly affected the surface morphology of the membrane. Both the increase in TOTM content and the decrease in the cooling rate effectively increased the mean pore size of ECTFE membranes, the former increased more but the mechanical properties of the membrane decreased significantly. The water recovery performance of the ECTFE membrane during the MC process increased with the increase in the mean pore size of the membrane. The condensation flow and water recovery rate of the membrane prepared with 20% TOTM were 1.71 kg·m^−2^·h^−1^ and 54.84%, respectively, which outperformed the commercial hydrophobic PVDF membrane in the MC, and showed the potential for the application of ECTFE membranes during the MC process.

## Figures and Tables

**Figure 1 membranes-12-00757-f001:**
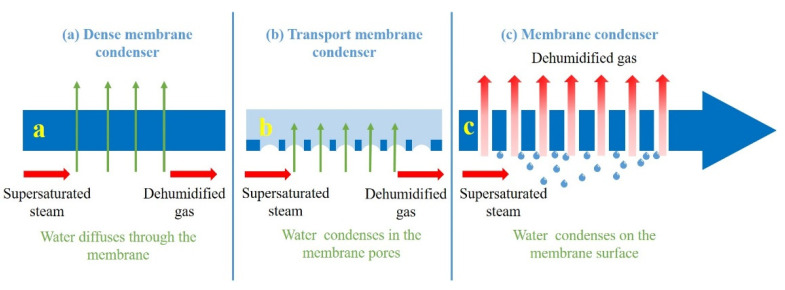
Dehydration mechanisms of different membranes: (**a**) DMC, (**b**) TMC, (**c**) MC [8].

**Figure 2 membranes-12-00757-f002:**
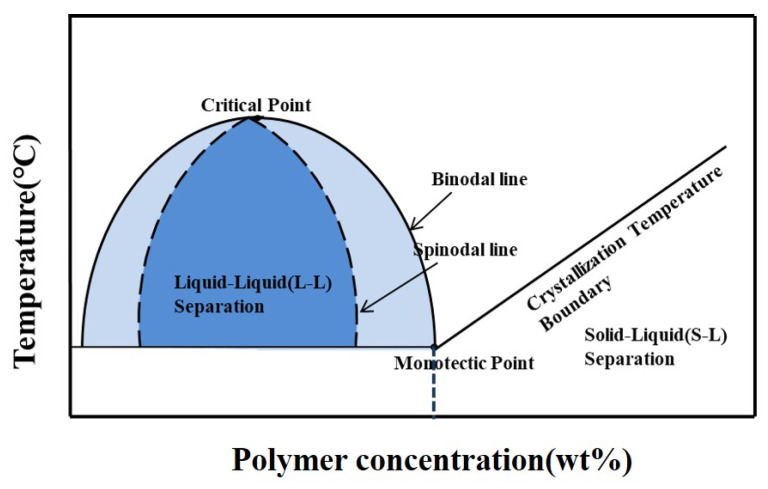
Phase diagram of PVDF/DBM system.

**Figure 3 membranes-12-00757-f003:**
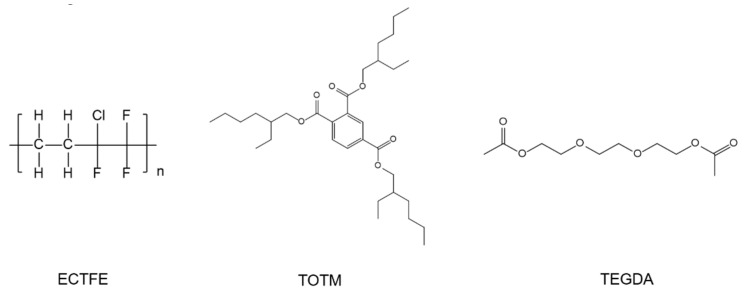
The molecular structure of ECTFE, TOTM and TEGDA.

**Figure 4 membranes-12-00757-f004:**
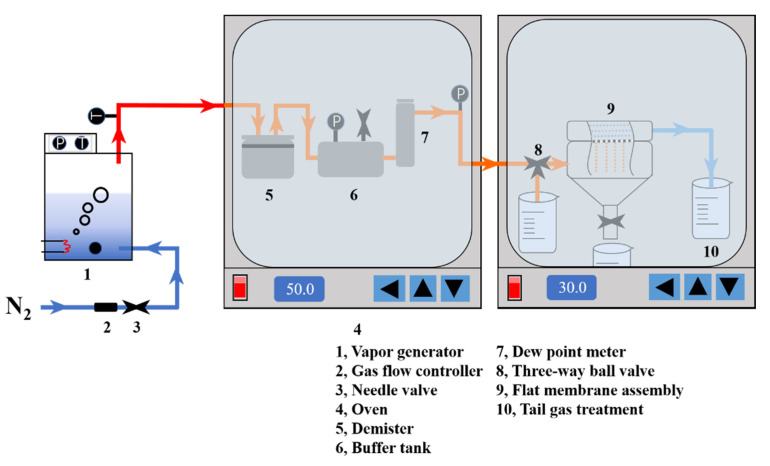
MC device based on the treatment of high-humidity gas.

**Figure 5 membranes-12-00757-f005:**
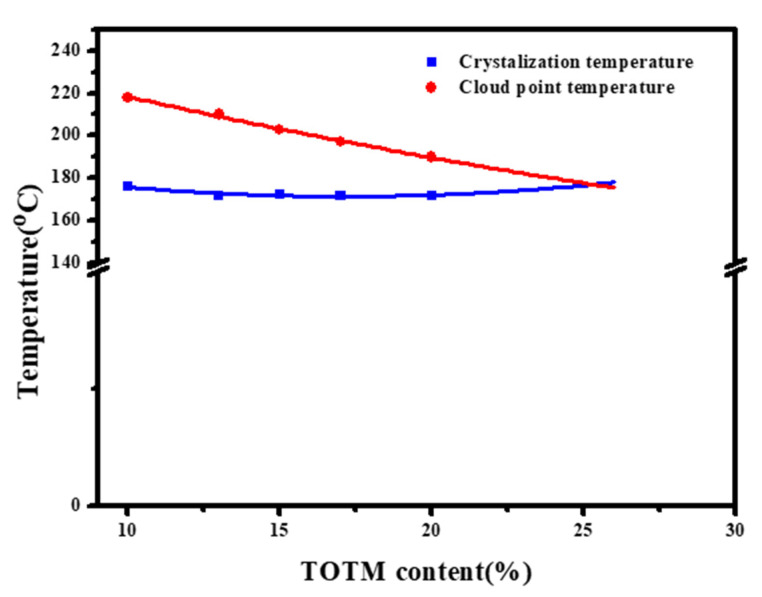
Thermodynamic phase diagram of ECTFE/(TEGDA:TOTM).

**Figure 6 membranes-12-00757-f006:**
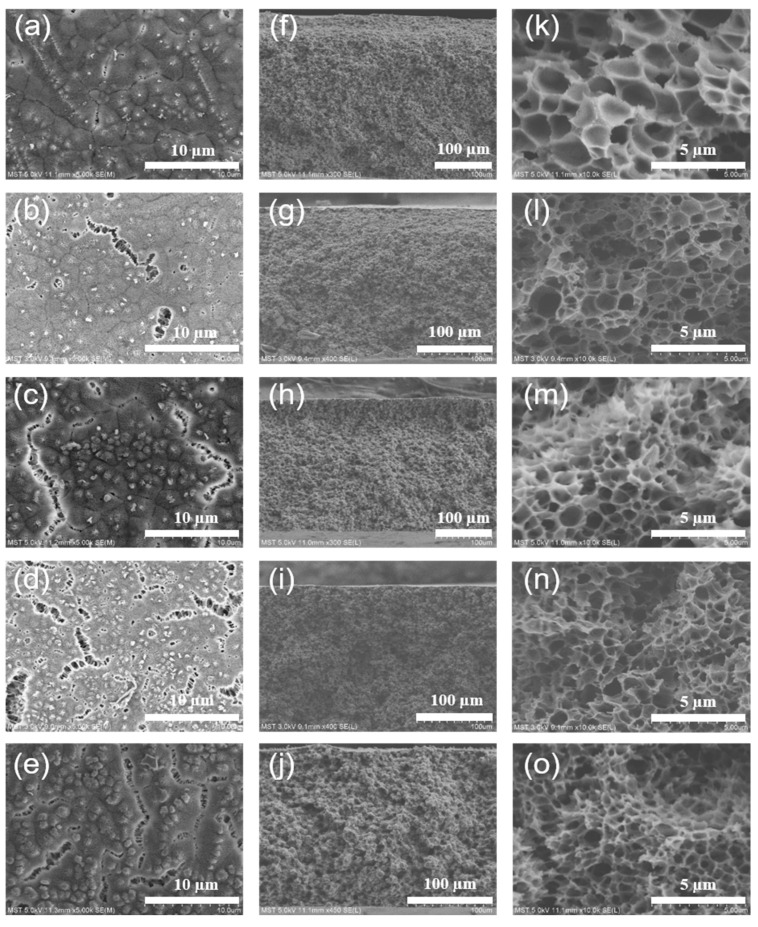
Surface SEM images of ECTFE membranes prepared with different TOTM contents: (**a**) 10%; (**b**) 13%; (**c**) 15%; (**d**) 17%; (**e**) 20%. Cross section SEM images of ECTFE membranes prepared with different TOTM content: (**f**,**k**) 10%; (**g**,**l**) 13%; (**h**,**m**) 15%; (**I**,**n**) 17%; (**j**,**o**) 20%.

**Figure 7 membranes-12-00757-f007:**
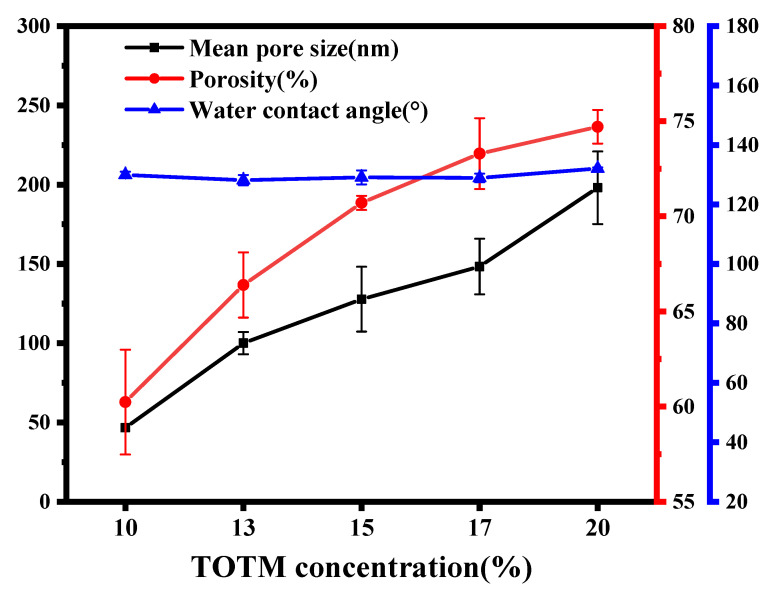
Mean pore size, porosity and water contact angle of ECTFE membranes prepared with different TOTM contents.

**Figure 8 membranes-12-00757-f008:**
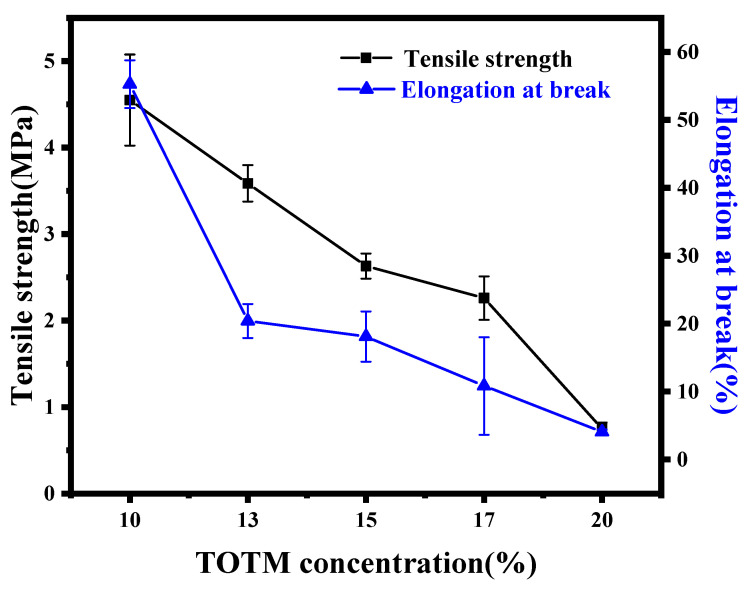
Mechanical properties of ECTFE membranes prepared with different TOTM contents.

**Figure 9 membranes-12-00757-f009:**
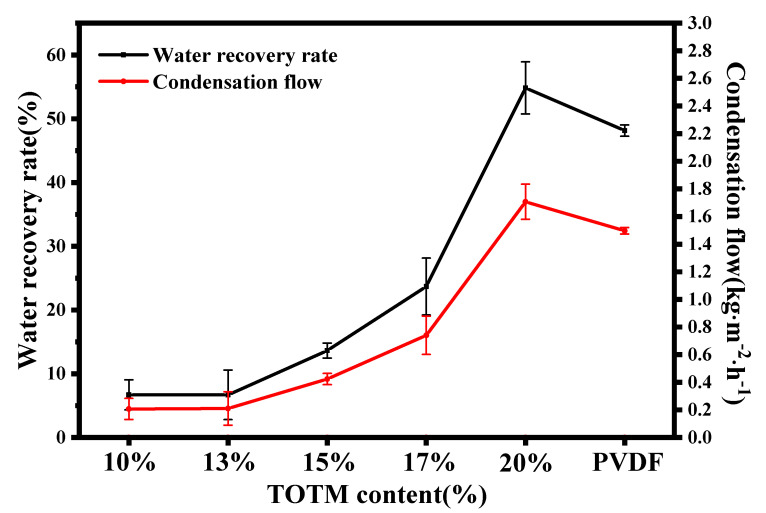
MC performance of ECTFE membranes prepared with different TOTM contents.

**Figure 10 membranes-12-00757-f010:**
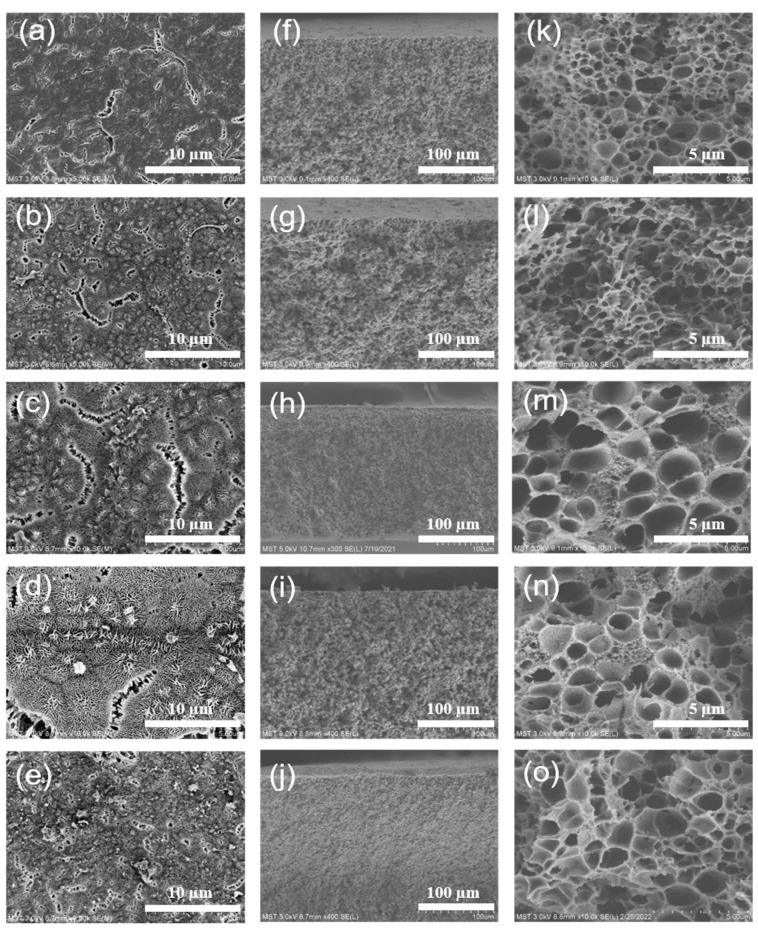
Surface SEM images of ECTFE membranes prepared at different cooling rate: (**a**) 0 °C; (**b**) 15 °C; (**c**) 30 °C; (**d**) 45 °C; (**e**) 60 °C. Cross section SEM images of ECTFE membranes prepared at different cooling rate: (**f**,**k**) 0 °C; (**g**,**l**) 15 °C; (**h**,**m**) 30 °C; (**i**,**n**) 45 °C; (**j**,**o**) 60 °C.3.2.2. Mean Pore Size, Porosity and Water Contact Angle.

**Figure 11 membranes-12-00757-f011:**
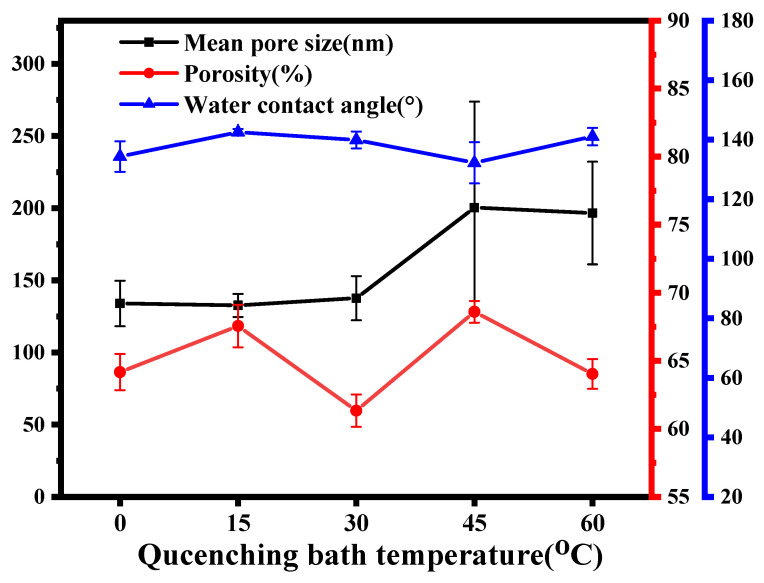
Mean pore size, porosity and water contact angle of ECTFE membranes prepared at different cooling rates.

**Figure 12 membranes-12-00757-f012:**
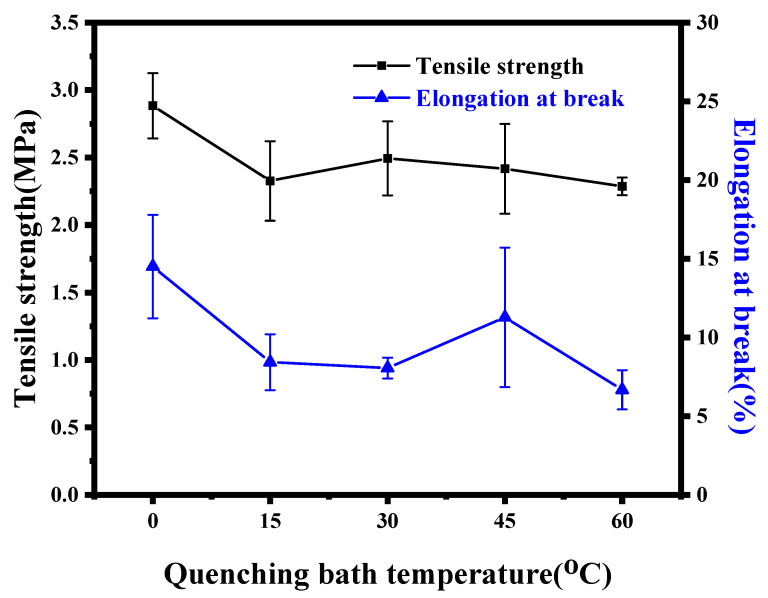
Mechanical properties of ECTFE membranes prepared at different cooling rates.

**Figure 13 membranes-12-00757-f013:**
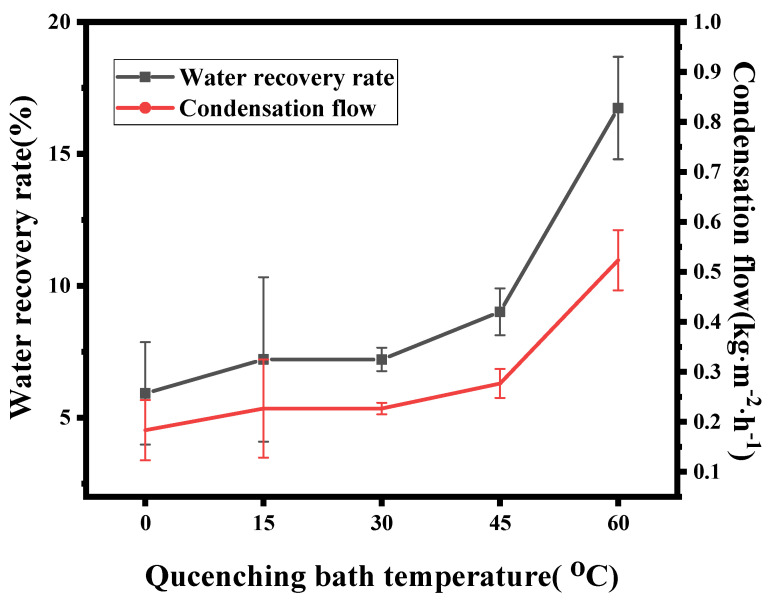
MC performance of ECTFE membranes prepared at different cooling rates.

**Table 1 membranes-12-00757-t001:** The parameters of MC process.

The Operating Conditions	Value
N_2_ flow rate (L/min)	0.5
Feed gas temperature (°C)	50
Feed gas relative humidity (%)	100
Cold sweep gas temperature ΔT (°C)	20
Operation time (h)	1
Membrane area (m^2^)	8.317 × 10^−4^

**Table 2 membranes-12-00757-t002:** Solubility parameters of ECTFE and diluents [22].

	*δ**_d_* (MPa ^1/2^)	*δ**_p_* (MPa ^1/2^)	*δ**_h_* (MPa ^1/2^)	*R* (MPa ^1/2^)
ECTFE	19.5	7.3	1.7	-
DEP	17.6	9.6	4.5	5.25
GTA	16.5	4.5	9.1	9.93
TOTM	16.66	8.55	6.03	8.54
TEGDA	16.45	2.14	9.78	11.36
DBM	16.5	6.1	7.2	8.23

**Table 3 membranes-12-00757-t003:** Literature data of membrane condenser.

Membrane	Membrane Area (m^2^)	Feed Gas Relative Humidity (%)	Feed Flow Rate (L⋅min^−^^1^)	Feed Gas Temperature (°C)	Water Recovery Rate (%)	Condensation Flow (kg·m^−2^·h^−1^)	Reference
Ceramic membrane-KRICT 100	0.00532	50,80	1–6	60–80	Not mentioned	0.5–11	Kim et al. [6]
Modified PVDF membrane M-40L	0.00252	60,95	1.0,2.0	50	5.7–18.85	0.15–0.35	Cao et al. [8]
Flat ECTFE membrane	0.004	100	0.076–0.38	55,65	35–55	Not mentioned	Drioli et al. [12]
Flat ECTFE membrane	0.001256	100	1.5	55	10–17	1.1–1.8	Pan et al. [28]
Ceramic membrane	0.0021	100	2	45–85	25–50	2–15	Wang et al. [31]
Flat ECTFE membrane (TOTM content is 15%)	0.000832	100	0.5	50	13.65	0.42	This work

## Data Availability

Not applicable.

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
