# Peer review of "ECTFE Membrane Fabrication Using Green Binary Diluents TEGDA/TOTM and Its Performance in Membrane Condenser"

_membranes, 2022, doi:10.3390/membranes12080757_

Round 1

Reviewer 1 Report

In this work the authors studied the effect of processing conditions (diluent composition and cooling rate) of ECTFE membranes, prepared by the TIPS method, on their porous structure and performance in a membrane condenser process. I recommend major revisions as detailed below.

The authors should indicate the meaning of ECTFE (i.e., poly(ethylene-chlorotrifluoroethylene)), in the abstract as well as the first time it appears in the introduction.

Also, in the introduction the authors should indicate the meaning of the initials DEP and GTA the first time they appear (line 89?). The same for ATBC and TOTM.

In the materials section 2.1 the authors should indicate the chemical formulas (CxHyOz etc) of ECTFE, TOTM and TEGDA.

In line 153, epsilon represents the open porosity – if there are closed pores kerosene will not enter them. Also related with this, the authors should discuss the viability and reliability of measuring porosity by immersion in kerosene considering that several of the samples (those with lower TOTM content) show clearly a closed pores structure - as can be seen in the SEM images. If they have closed pores how can kerosene penetrated those pores? The authors should comment on this.

On line 149 says that "The pore size was measured by a membrane pore size distribution apparatus". More details should be provided. According to Figure 5, the mean pore size varies between 50 and 200 nm. However, looking at the SEM images much larger pores can be seen and the variation of pore size in the SEM images (pore size decreases with TOTM content) is opposite to the one represented in Figure 5 (mean pore size increases with TOTM content). The authors should clarify these apparent contradictions. Are there pores in two different lengths scales?

In Figure 5, the legend of water contact angle is in green but the results are in blue. Please correct.

In line 277 should be Fig. 4 and not Fig. 6.

In line 282, do the authors really mean “further reducing”? Or do they mean “further increasing”? “Further reducing” seems illogical in the context of this and the previous sentences.

The article includes too many self-citations: at least 8 out of 20, that is 40 %, namely refs numbers 3, 4, 5, 6, 8, 13, 14, 15. Authors should reduce the number of self-citations (to a maximum of 5 most important) and increase the number of citations to other people´s work.

Author Response

Dear Reviewer, please see the attachment, thank you very much.

Reviewer 2 Report

The manuscript entitled “ECTFE membrane fabrication using green binary diluents TEGDA/TOTM and its performance in membrane condenser” describes the application of binary solvent mixtures of TEGDA/TOTM to prepare membrane condenser materials made from ECTFE using TIPS technique. The manuscript can be accepted in Membranes (MDPI) following these specific comments:

1.     Figure 1c: the labelling of “dehumidified gas” is overlapped with “(c) membrane condenser”, please fix this.

2.     2.3 ECTFE membrane preparation, line 143: Why 20% ECTFE was chosen? Why there was no variation of ECTFE concentration in this study?

3.     Table 1: N2 flow rate, please subscript the number.

4.     Line 213-214: what is a wide L-L  phase separation region? This needs some citations to better explain the concept. Maybe some illustrations to enhance the comprehension of the reader rather than just some texts.

5.     Line 224 – 226: it was mentioned that TOTM is a good solvent. A table with the Hansen solubility parameters of TOTM, TEGDA, DEP and GTA will be important to verify the claim.

6.      Line 263: “around 130 o” please superscript the “o”.

7.     Figure 7 and Figure 11: there is a formatting error in the condensation flow unit, please fix them.

8. A table containing the previous published results can be added to compare the performance of different membrane condenser with this current work.

Author Response

(The authors gave the same response as above.)

Round 2

Reviewer 1 Report

The authors have carefully addressed all the reviewer comments. 

I recommend publication in present form.

Reviewer 2 Report

The authors have made necessary changes and I recommend the manuscript to be accepted for publication.